# Responses of candidate intensity measures to different mental and motor load levels using upper limb exergames in typically developing children and adolescents

**Gaizka Goikoetxea-Sotelo**[1,2,3]*, **Livia Rätzo**[1,2,3], **Hubertus J. A. van Hedel**[1,2]

**1** Swiss Children's Rehab, University Children's Hospital Zurich, University of Zurich, Affoltern am Albis, Switzerland, **2** Children's Research Center, University Children's Hospital Zurich, University of Zurich, Zurich, Switzerland, **3** Department of Health Sciences and Technology, ETH Zurich, Zurich, Switzerland.

\* gaizka.goikoetxea@kispi.uzh.ch

## Abstract

### Background

In (pediatric) neurorehabilitation, high-intensity therapy is well-recognized for enhancing rehabilitative outcomes. However, measures that consider the different aspects of therapy intensity are lacking. Therefore, a better understanding of how to measure therapy intensity during upper-limb neurorehabilitation is needed.

### Objectives

To investigate the response of heart rate variability (HRV), skin conductance (SC), activity counts and movement repetitions normalized for the maximal capacity (%ACmax and %MOVmax, respectively), and the NASA-TLX scale to different mental and motor intensity levels of two self-developed upper limb exergames in typically developing children. We also investigate the effects of age on the responses of the measures.

### Methods

In this cross-sectional study, participants engaged in one mental and one motor exergame. For each exergame, they played three personalized intensity levels ("very easy," "challenging," and "very difficult"), each lasting three minutes. We analyzed the responses of all measures for each intensity level and exergame.

### Results

21 children and adolescents (9 females) aged 5.2 to 17.9 years participated in the study. HRV, %ACmax, and %MOVmax responded to increased motor intensity. SC did not respond to increases in mental or motor intensity levels. The NASA-TLX

License, which permits unrestricted use, distribution, and reproduction in any medium, provided the original author and source are credited.

**Data availability statement:** All files are available from the Harvard Dataverse repository (https://doi.org/10.7910/DVN/AVKHG2).

**Funding:** This work is supported from grants from the J&K Wonderland Foundation and the Anna-Müller Grocholski Foundation. The funders had no role in study design, data collection and analysis, decision to publish, or preparation of the manuscript.

**Competing interests:** The authors have declared that no competing interests exist.

**Abbreviations:** HRV, Heart rate variability; NASA-TLX, National aeronautics and space administration task load index questionnaire; IMU, Inertial measurement unit; ROM, Range of motion; SC, Skin conductance; AC, Activity counts; MOV, Movement repetitions; ACmax, Maximal activity count capacity; MOVmax, Maximal movement repetition capacity; %ACmax, Relative activity counts; %MOVmax, Relative movement repetitions

responded to increases in motor intensity levels but only partially to increases in mental intensity. Finally, age showed significant effects only on %MOVmax.

## Conclusion

Changes in mental intensity were more challenging to capture than changes in motor intensity. As each measure responded to different aspects of therapy intensity, a combination of measures, e.g., HRV, %ACmax, and NASA-TLX effort, might be the best strategy for assessing therapy intensity multidimensionally. Although the measures hold considerable potential, future studies should determine the responses of the measures and their psychometric properties in the target group.

---

## 1 Introduction

In (pediatric) neurorehabilitation, individuals with congenital or acquired neurological diagnoses receive specialized treatments. These treatments, which aim to improve patients' independence and quality of life [1,2], are based on motor learning theories [3]. Therapists adapt the therapies individually to the patient's circumstances, abilities, and goals. Treatment effects are evaluated using standardized rehabilitative outcome measures. Improved outcomes have been consistently associated with higher therapy intensities [4–8], which highlights the importance of intensity in neurorehabilitation.

Although intensity is one of the most critical factors affecting rehabilitation outcomes for upper limb motor learning-based neurorehabilitation [9], there are no valid and universal measures to quantify it. On an individual level, valid and universal intensity measures would quantify the actual intensity level at which a patient is working, enabling a better personalization of therapy. At a higher level, they might allow comparing the effectiveness of different interventions. For such a comparison, the interventions must be dose-matched. As intensity is a major component of dose, valid intensity measures are needed to ascertain that the dose of the therapies is matched, and, therefore, conclude that the differences in effectiveness derive from the type of intervention.

Because of the lack of valid and universal intensity measures, therapy intensity is usually equated with time spent in therapy. However, time spent in therapy is a very poor proxy for what a patient does during therapy [10], and the concept of therapy intensity in rehabilitation extends clearly beyond it. Therapy intensity has been referred to as "the amount of mental and motor work put forth by a patient" [11]. For instance, if we take the closing of a jacket's buttons as an example (often trained during occupational therapy), the intensity could be increased by increasing the number of repetitions (i.e., closing more buttons), which would merely increase the motor work. However, a therapist could also ask the patient to close smaller buttons that are more difficult to hold. If we would measure only the time, it would remain unaltered, e.g., both therapy exercises lasted 10 minutes. However, closing smaller buttons becomes a more fine-skilled task requiring more precise motor planning and

concentration, which would increase the motor and mental work. Thus, it becomes evident that time spent in therapy is an inadequate estimate of therapy intensity. An integral understanding of therapy intensity in neurorehabilitation necessitates assessments that evaluate both the motor and mental dimensions of the patient's work.

With the quantification of intensity in mind, several measures have been developed and applied in the context of upper limb neurorehabilitation. For instance, measures reflecting the amount of movement, such as activity counts (AC) or the number of movement repetitions (MOV), have been suggested to be better representatives of what a patient does during therapy than time spent in therapy [12,13]. Although we agree that these are more precise than time spent in therapy, such measures cannot capture other motor and mental work aspects, such as movement complexity. In addition, AC and MOV are absolute measures and represent the amount of movement patients perform during a therapy session rather than how hard they work. To use AC and MOV as intensity measures and tailor them for therapy, we would require additional information, such as maximum capacity values, e.g., the highest number of counts (ACmax) or movement repetitions (MOVmax) a patient can achieve for a specific exercise and time period. Such maximum capacity values would enable us to determine the relative intensity level (%ACmax or %MOVmax) at which the patient is working.

Self-reported effort scales such as the adapted Borg scale are relative measures that reflect exercise intensity at a more complex and multidimensional level [14,15]. However, the Borg scale represents "how hard one feels their body is working", and its interpretation may neglect the mental work required to meet the demands of more complex tasks. The NASA-TLX questionnaire might overcome this limitation as it offers a comprehensive view across six dimensions, encompassing physical and mental workload [16]. Furthermore, it may be helpful in clinical settings due to its simplicity, sensitivity, reliability, and validity [17,18] and because previous studies highlighted the questionnaire's ability to differentiate difficulty levels and specific task demands [19,20]. Nevertheless, the utility of perceived exertion questionnaires for online measurement of therapy intensity seems limited, as repeatedly asking the patient to rate perceived exertion could become tedious and disrupt the flow of the session, reducing engagement and potentially compromising the therapy. Furthermore, the subjective nature of perceived exertion ratings introduces variability and subjectivity, challenging the interpretation and comparison across sessions or individuals.

Physiological measures such as heart rate variability (HRV) and skin conductance (SC) are considered objective and easy-to-use measures of heart-brain interaction and central nervous system modulation. Heart rate variability tracks the variation in time intervals between consecutive heartbeats, offering insights into the dynamic balance and flexibility of the autonomic nervous system [21]. Elevated HRV, which is often observed in physiologically relaxed or non-stressful situations, typically indicates greater adaptability and regulatory capacity of the CNS over physiological processes. In contrast, reduced HRV, commonly seen in physically or cognitively demanding or stressful situations, may suggest a less adaptable and potentially compromised neural regulatory system. Skin conductance measures the skin's electrical conductivity, indicating the autonomic nervous system arousal in response to emotional, cognitive, or physiological stimuli [22]. Increased or high skin conductance often manifests in situations with increased sympathetic arousal, while reduced or low skin conductance is usually seen in relaxed or non-stressful situations. Due to these characteristics, studies analyzing the effects of increments in mental and motor task difficulty in healthy adolescents and adults have found changes in HRV and SC profiles [23–25]. This responsiveness suggests their potential as measures of intensity in pediatric upper limb neurorehabilitation. However, we are unaware of their ability to respond to changes in therapy intensity, particularly in children.

Indeed, children's anatomy, physiology, and cognition continue to develop into adulthood [26], and these stages of development could influence the measures' responses. For instance, when measured during rest, HRV and SC decrease with increasing age [27,28]. However, whether this effect is maintained when measuring during active periods is unknown. As additional factors such as diagnosis and medication could impact physiological responses, and cognitive impairments may affect subjective responses, we first investigated the responses of the measures to changes in mental and motor intensity in typically developing children and adolescents.

 

While each measure provides a unique perspective on therapy intensity, the complexity and multidimensional nature of intensity pose a challenge for any single measure to encompass the entire spectrum of therapeutic goals and activities. To understand how we could best capture the multidimensionality of intensity in upper limb neurorehabilitation, we investigated how HRV, SC, %ACmax, %MOVmax, and the NASA-TLX responded to personalized motor and mental intensity levels of upper limb exergames. We hypothesized that 1) HRV would decrease, while SC and the NASA-TLX would increase with increasing mental and motor load, 2) %ACmax and %MOVmax would increase with increasing motor load and would show no effect with increasing mental load, and 3) elderly children would have lower HRV and SC compared to younger children, and age would not affect %ACmax, %MOVmax, and the NASA-TLX.

## 2 Methods

### 2.1 Participants

The goal was to recruit 21 typically developing children and adolescents evenly distributed across three age groups: 5–8, 9–12, and 13–17 years. The age groups were based on Jean Piaget's theories of cognitive development [29], covering early childhood, late childhood, and the final stage of cognitive development. The inclusion criteria were: age 5–17 years; no current neurological, musculoskeletal, psychological, or cardiovascular diagnoses; ability to understand and follow simple verbal instructions; no visual impairments that interfere with computer play; intact skin in the hypothenar eminence of the dominant hand, neck, and trunk; and ability to communicate pain and discomfort.

We informed eligible participants and their legal representative(s) verbally about the study; for those aged ten years and older, also in writing. Assent was mandatory from all participants. We also obtained written consent from one legal representative and the children aged 14 years and older. The Cantonal Ethics Committee of Zurich clarified the study's responsibility (BASEC no. Req-2023–00206). The study adhered to the Declaration of Helsinki and good clinical practice guidelines.

### 2.2 Procedures

**2.2.1 General procedures.** The study was conducted at the Swiss Children's Rehab of the University Children's Hospital Zurich from the 23.03.2023 to the 12.05.2023. During 60 minutes, the participants engaged in two custom-made exergames on the Myro® (Tyromotion, Graz, Austria; see Fig 1). We designed one exergame specifically to investigate the responses of the various candidate intensity measures to different levels of motor load, while the other game aimed to examine responses to mental load levels. The participants played each exergame with their dominant hand at three individualized intensity or task difficulty levels ("very easy", "challenging", and "very difficult"). We randomized the order of the exergames and intensity levels to account for fatigue.

After welcoming the participants, we explained the procedures and protocols (Fig 2). Subsequently, a therapist attached the Polar chest strap to measure the heart rate, the electrodes to measure SC, and the IMU to measure activity counts. All devices were switched on simultaneously. For the baseline measurement, we measured the heart rate and SC for 3 minutes while the participant was at rest. To ensure that the participants could reach each playable corner, we adapted the workspace to the individual range of motion (ROM) and the distance between two consecutive targets remained between 60% and 80% of the active ROM. The protocol comprised three phases for each exergame, i.e., familiarization, calibration, and measurement. In the familiarization phase, we explained the task and goal of the exergame to the participants while they gained confidence by playing it for 2 minutes. During the first familiarization, we explained the NASA-TLX, clarified comprehension to reduce measurement error, and encouraged the participants to consider intermediate responses instead of limiting them to extremes [30]. In the calibration phase, we performed a maximal capacity test, which we used to personalize the intensity levels according to the participant's abilities. During the measurement, the participants played each intensity level for 3 minutes. The therapists provided standardized input on game duration at specific times and were

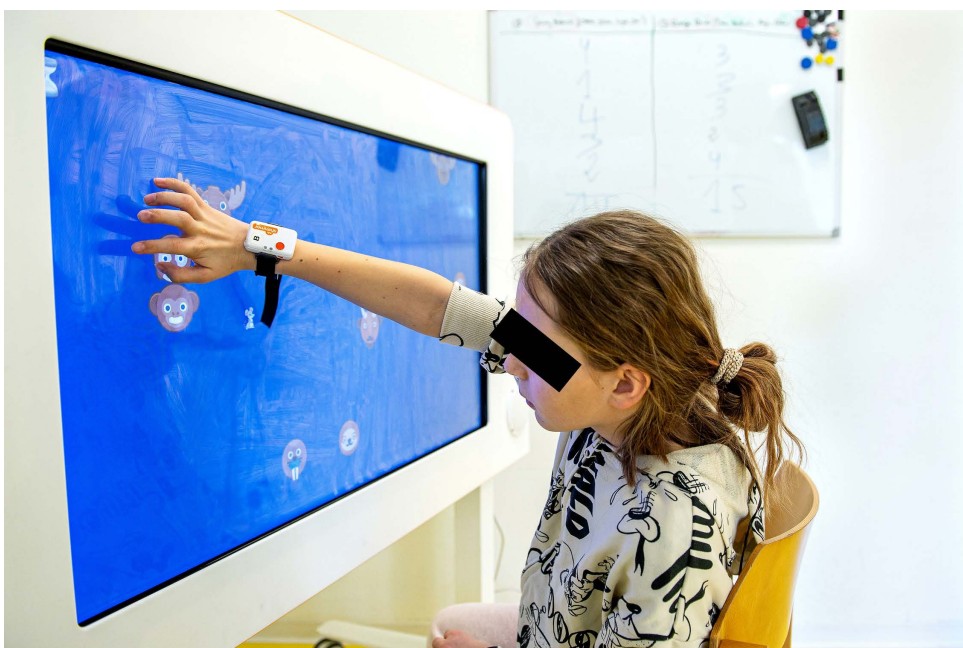

**Fig 1. A child plays the custom-made mental exergame using the Myro® while wearing a Shimmer® inertial measurement unit (IMU) at the wrist.** The Myro® is a device with a 941x529 mm touchscreen. The therapist can adapt the device to the patient's needs by adjusting the angulation, height, and work surface. The device responds to motion, pressure, and pulling, and patients can steer the games using their hands or objects that require different grasps. In therapy, the Myro® enables the training of gross and fine motor skills through video gaming. For this study, children steered the game using their dominant hand and fingers.

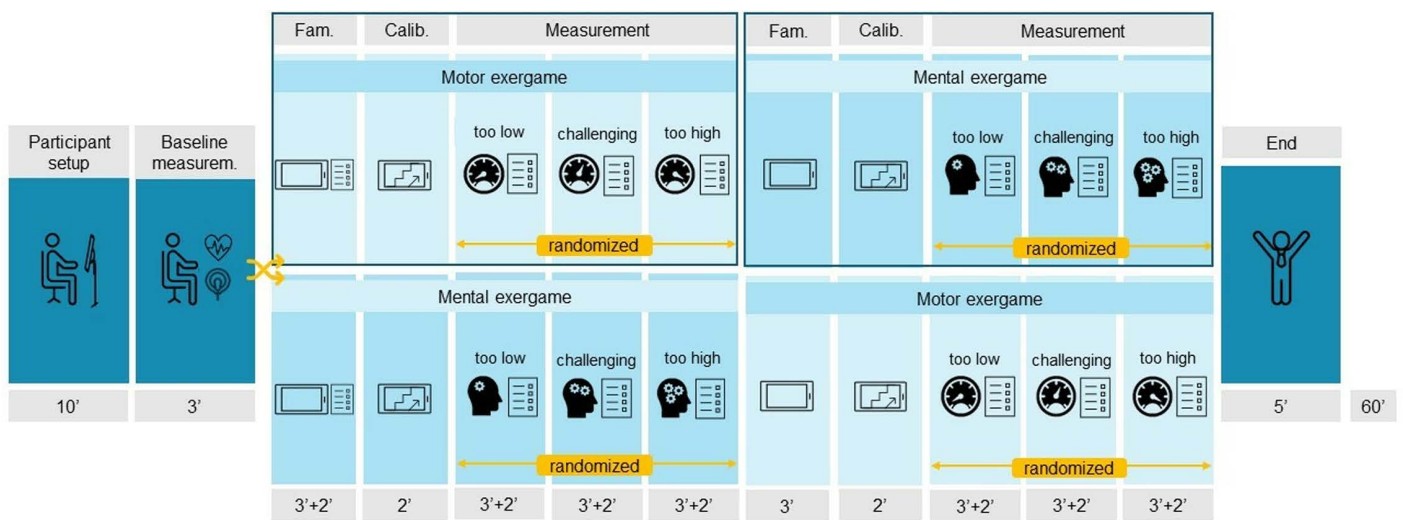

**Fig 2. Study protocol.** After the participant setup and a baseline measurement, the participant played two exergames in a randomized order. Each exergame consisted of a familiarization, a calibration, and a measurement, where the participant played each exergame at three intensity levels (i.e., "very easy", "challenging", and "very difficult") in a randomized order. While playing, we measured the responses of the various intensity measures to each intensity level. After each intensity level, the participant answered the NASA-TLX questionnaire.

allowed to provide motivational input to the participant. During each level, a therapist noted the participant's behavior. Between the levels, the participant completed the NASA-TLX questionnaire.

## 2.3 Custom-designed exergames

We developed two exergames (the mental and the motor exergame) that allowed us to create controlled and replicable environments. Differently from off-the-shelf exergames [31], our custom-designed exergames enabled us to tailor task load levels to the individual capacities of the participants and to increase mental and motor load independently. These characteristics facilitated the creation of a more structured and systematic approach, which helped us interpret the results better. We uploaded the games to the Harvard Dataverse repository (https://doi.org/10.7910/DVN/AVKHG2).

 **2.3.1 Mental exergame.** The mental exergame is a visual search task in which participants have six seconds to locate and hit a target among distractors (Fig 3). The number of distractors depends on the intensity level (i.e., the higher the intensity, the more the distractors) and on the ability of the individual participant. After familiarizing themselves with the game,

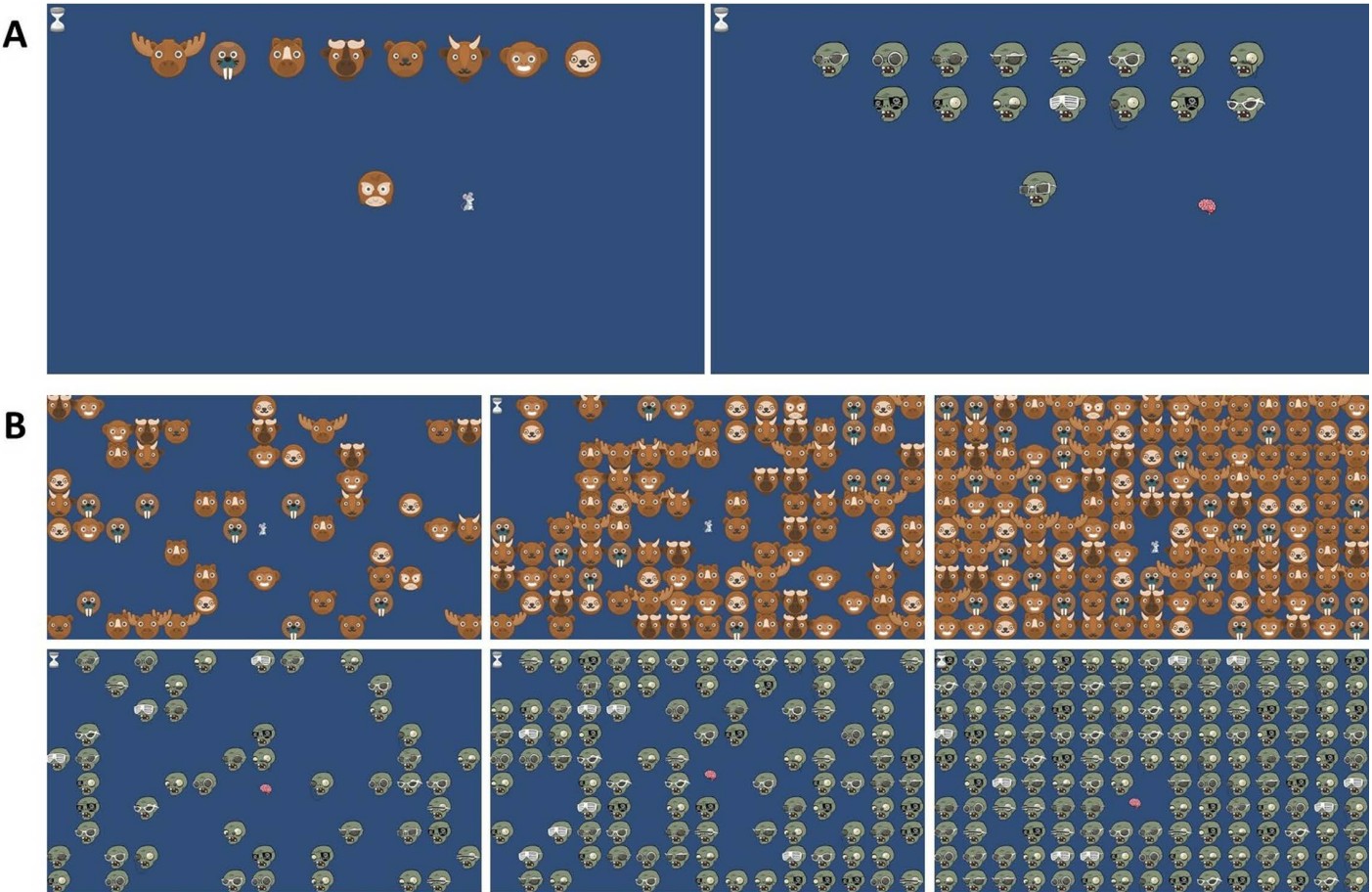

**Fig 3. Mental exergame with the «owl with a brown beak» and «zombie with square glasses» targets. (A)** Start screens before familiarization, calibration, and exergame with the target object (bottom) and its distractors (top). The start screen allowed the participant to memorize the target better. **(B)** Examples of three intensity levels for the «owl with a brown beak» and «zombie with square glasses»: a "very easy" intensity level (50% according to the calibrated output, left), a "challenging" level (100% of calibrated output, middle), and a "very high" intensity level (150% of calibrated output, right). The three intensity levels are compared here to visualize how the number of distractors changes. The participants performed the exergames on the large Myro® screen, where the targets and distractors were well recognizable.

the participants underwent a calibration procedure consisting of an incremental test. As in the exergame, the test involved locating and hitting a target among the distractors within six seconds. At the start of the test, eight distractors and one correct target appeared on the screen. If the participant found the target within six seconds, the remaining time until the end of the six seconds was granted as rest. After each successful trial, four additional distractors came into play. If the participant did not find the target within six seconds, the trial was repeated with the same number of distractors. The calibration continued until the participant failed to find the target within the time limit three times consecutively. While calibrating, it was possible to reach the maximum score. In this case, we selected another target that differed from its distractors in fewer features to challenge these participants further. A higher similarity between target and distractors requires an increased focus and search efficiency, making the detection more difficult [32]. After selecting the new distractor, the participant repeated the familiarization and calibration procedures until a valid calibration was achieved. We selected the number of distractors from the last level where the participant managed to hit the target in time to determine the three intensity levels.

Defining mental intensity levels posed a challenge due to the limited literature on thresholds for visual search tasks. Drawing on game design theories and the expertise of the Swiss Children's Rehab research team, we set the mental intensity levels at 50%, 100%, and 150% of the calibration output. The aim was to achieve similar success rates as the ones suggested by Aponte and colleagues [33], who categorized performance thresholds below 30% as "bad," 30–60% as "average," and above 60% as "good" performance. We adjusted the percentages for child-friendliness and game design based on pilot testing.

**2.3.2 Motor exergame.** The motor exergame requires gross motor arm movements to hit a balloon within a given time (Fig 4). The time depends on the intensity level (i.e., the higher the intensity, the shorter the time), which is based on the calibration test. After the familiarization phase, the participants performed the calibration, which consisted of a 30-second test. During the test, a balloon appeared on the screen and remained there until the participant hit it. The participant had to hit the balloon as quickly as possible. When the participant hit the balloon, a new one appeared in a new position until the 30 seconds were up. We selected the mean speed needed to hit the balloons to determine the intensity levels. The mean speed considered the distance and time from the previous target position to the next and was calculated using Equation 1.

$$Mean\ speed\ [m/s] = \frac{\sum_1^N \frac{distance_{new-last\ target\ position}}{time_{target\ appearance\ -touch}}}{N}$$

Equation 1: Calculation of mean speed for motor exergame
(N: Total number of balloons that appeared during the calibration)

During gameplay, the participants had to hit the target within the given time. The time the target remained displayed depended on the individual calibration procedure and the intensity level (i.e., the higher the intensity, the shorter the time

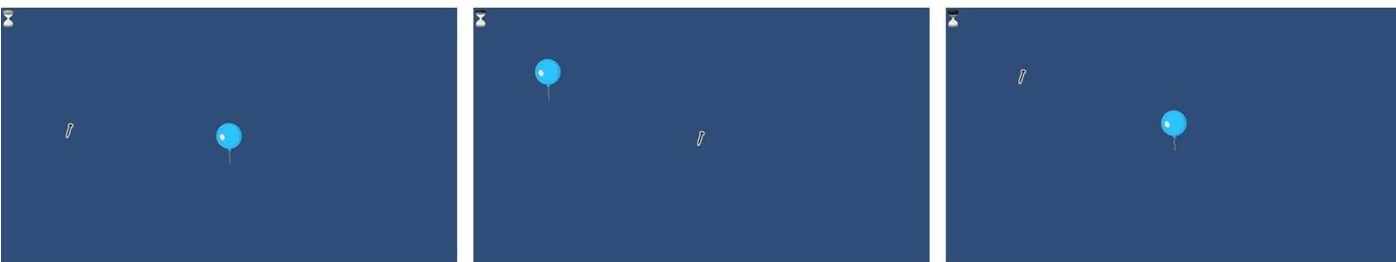

**Fig 4. Motor exergame.** The participants had to hit the balloon with the nail. Once the balloon was hit, it reappeared in another position. The distance between the occurrence of two consecutive balloons is always between 60-80% of the participant's range of motion. The time of reappearance depended on the calibration procedure and the intensity level. (N: Total number of balloons that appeared during the calibration).

window). After the participant hit the target, it disappeared and reappeared at a new location on the screen (randomly selected by the software). If the participant hit the target within the allotted time, the remaining time until the new target appeared was granted as rest. Successful target hits triggered positive auditory feedback, whereas negative auditory feedback was given if the time elapsed without a hit.

Also for the motor game, defining intensity levels posed a challenge due to the limited literature on specific threshold values. Many factors influence a game's difficulty, making the correct adjustment of exergame intensities to a participant's ability quite complex. Besides game design theories and the expertise of the Swiss Children's Rehab research team, the work of Zimmerli and collaborators on balancing mechanisms in upper extremity rehabilitation applications guided this process [34]. The authors implemented Fitts' Law, which adapts difficulty based on movement time, target size, and distance into feedback systems to reach targets from a starting point within a given time. They applied easy, balanced, and hard levels based on the estimated time to reach the target, i.e., they increased the difficulty by shortening the available time window. We established similar intensity levels for the motor exergame but adjusted the percentage following extensive testing with our user group. We selected 0% of the mean calibration speed for the "very easy," 70% for the "challenging", and 100% for the "very difficult" intensity level.

## 2.4 Success rate and candidate intensity measures, including data processing

**2.4.1 Game-based control variable: Success rate.** The success rate describes the percentage of correct interactions with the exergame [i.e., 100· (correct interactions/incorrect interactions)]. The game software recorded the participant's interactions with the exergame and calculated the percentage of successful interactions for each intensity level. The success rate is not an intensity measure but a control variable that helps us to understand whether we had set the intensity levels as intended. For the mental condition we expected success rates of around 100% for the "very easy" level with little effort, 70% for the "challenging" level with high effort, and 50% for the "very difficult" level with maximum effort. In contrast, for the motor condition, we aimed for success rates of 100% for the "very easy" intensity level without much effort, nearly 100% for the "challenging" intensity level with considerable effort, and around 70% for the "very difficult" intensity level with extreme effort.

**2.4.2 Physiological measures.**

**Heart rate variability** For data collection, participants wore a chest strap with the Polar H10 Heart Rate Sensor (Polar Electro Oy, Kempele, Finland) tightened under the pectoral muscles (Fig 5A). The sensor recorded the time intervals between heartbeats (raw R to R intervals) in milliseconds at a sample frequency of 1000 Hz and stored them in the Elite HRV application (Asheville, USA).

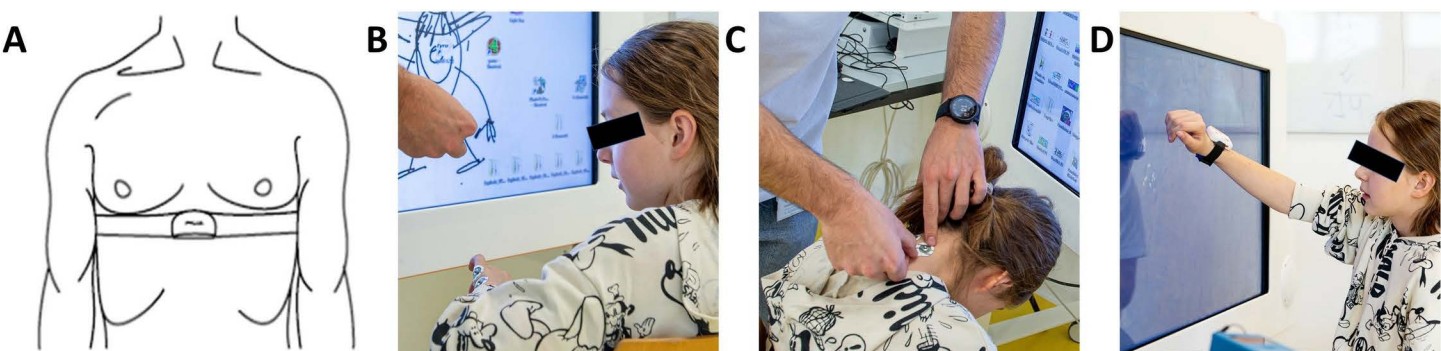

**Fig 5. Participant setup. (A)** A chest strap with a Polar H10 Heart Rate Sensor tied tightly under the pectoral muscles measures heart rate variability. We assessed skin conductance using electrodes positioned on **(B)** the hypothenar side of the non-dominant hand and **(C)** the neck para-medial below the hairline. **(D)** An IMU sensor placed dorsally around the wrist of the dominant hand to measure activity counts. The blue box on the table is the MentalBioScreen K3 device for measuring skin conductance.

 

In our study, we employed the time-domain variable RMSSD (root mean square of the successive differences) to quantify HRV. We processed the raw data with Kubios HRV Standard 3.5.0 software (University of Eastern Finland, Kuopio, Finland OR Kubios Oy LC, Finland). We used MATLAB® Runtime R2021a (MathWorks, Natick, USA) for further data mining. We manually set a filter-based threshold to correct up to 1% of erroneous beats by interpolating R-R intervals [35–37]. We chose manual filtering over «automatic» [38] and «strong» filters [37], as these filters seem to overgenerously remove minimum and maximum R-R intervals, automatically reducing HRV and thus biasing the results. Kubios automatically conducted subsequent signal processing steps, including signal smoothing (removal of frequencies below 0.04 Hz), cubic spline interpolation, and 4-Hz resampling [39]. We divided the time-domain data by the average R-R interval to normalize the evenly sampled time series [40,41]. This normalization addressed the nonlinear inverse relationship between R-R intervals and heart rate [42], preventing heart rate differences from affecting the comparisons. Equation 2 shows the formula used to calculate the RMSSD (average beat-to-beat variance).

$$RMSSD\,[ms] = \sqrt{\frac{1}{N-1}\sum_{i=1}^{N-1}\left(RR_{i+1} - \overline{RR_i}\right)^2}$$

Equation 2: Calculation of RMSSD for heart rate variability
(RR: Time difference between two successive heartbeats; N: Number of heartbeats)

**Skin conductance**   In this study, the MentalBioScreen K3 device (Porta Bio Screen GmbH, Berlin, Germany) measured the skin conductance in microsiemens (µS). We placed two pairs of Vivomed H5 0310 electrodes on the hypothenar side of the non-dominant hand (Fig 5B) and two on the neck, below the hairline (Fig 5C). Using MATLAB®, we calculated the mean SC over the last two minutes of the 3-minute recordings for each intensity level, separately for the neck and hand, to account for delayed physiological responses and the participant's initial nervousness.

### 2.4.3 Movement-based measures.

**Activity counts**   We used a Shimmer3® IMU (Shimmer Research Ltd, Dublin, Ireland) to assess the activity counts of each intensity level. We attached the sensor dorsally around the wrist of the participant's dominant hand (Fig 5D). It provides raw, three-dimensional acceleration data. We used an open-source MATLAB® script that band-pass filtered, resampled, and summed raw 3-dimensional acceleration data [43]. We calculated the activity counts per minute (AC/min) by summating 60 consecutive values [44]. We set the AC/min output from the 30-second motor calibration test as the maximal AC/min capacity (ACmax). We normalized the data from each intensity level to the maximal capacity to get the actual AC intensity measure, i.e., the percentage of the maximal AC capacity (%ACmax).

**Movement repetitions**   The number of MOV, as outputted by the exergames, indicates each participant´s interaction with the exergames. We summed the number of correct and incorrect interactions obtained from the exergame output to compute the total number of MOV. We defined the maximal movement repetition capacity (MOVmax) as the number of MOV from the 30-second motor calibration test, correcting it for time difference. We normalized the data of each intensity level to this number to get the actual MOV intensity measure, i.e., the percentage of the maximal number of movement repetitions capacity (%MOVmax).

In some cases, where %ACmax and %MOVmax slightly exceeded the 100% threshold for the "very difficult" intensity level, we scaled the data to 100% for a better interpretation and visual representation of the results.

### 2.4.4 Participant-reported measure: NASA-TLX questionnaire.
The NASA-TLX is a self-reported assessment that rates perceived workload across six dimensions: mental demand, physical demand, temporal demand, performance, effort, and frustration. The NASA-TLX is valid and reliable in both adults [17,18] and children [30]. We used the German version of the NASA-TLX questionnaire for children Harvard (the exact document in German and English can be found under https://doi.org/10.7910/DVN/AVKHG2). In addition, we translated it into Swiss German to adapt to the local culture

and language, maintaining content alignment. We used the Swiss German version only when the participants could not understand the standard German version. To address numerical literacy concerns, the participants used a numberless LEGO® scale, with a hidden scale ranging from 0 to 100 in increments of 5, and moved a slider to reflect their subjective experience. Consistent with Laurie-Rose et al., we omitted subscale weighting for faster completion and reduced cognitive demand during breaks [30]. The data processing involved subtracting the score of the performance subscale from 100 because of reverse scoring. We summed the subscale scores into an overall workload score, with higher values indicating an increased workload. We considered the overall score and the mental, physical, and effort subscale scores for the analysis. We selected these specific subscales in addition to the overall score, as we expected that the mental subscale would help us better understand the specific increases in mental load, the physical subscale would help us better understand the specific increases in motor load, and the effort subscale would serve as a combination.

### 2.5 Statistical analysis

We performed the statistical analysis using R 4.3.2 (RStudio Inc., Boston, USA). We fitted a linear mixed-effects regression model to the data [45]. We used the intensity outcomes as dependent variables, age as an independent variable, condition (motor or mental) and intensity level ("very easy," "challenging," and "very difficult") as independent variables with interaction, and the participant ID as a random intercept. We set the alpha level at 0.05; however, because this is an exploratory study with a small sample size and we expected considerable heterogeneity in the data, we decided to discuss tendencies too ($0.05 < p \leq 0.10$). We calculated the 95% confidence intervals for the fixed effects. We analyzed the following model for each intensity measure separately:

$$Outcome \sim Age + Condition * Intensity\ Level + (1|Participant)$$

## 3 Results

Twenty-one typically developing children and adolescents (9 females), with a mean age of 10.95 (SD 3.8) years (range 5.2 to 17.9 years), participated. No data loss occurred.

Table 1 shows the results of the regression models. The intercept of the regression model shows the results for the average-aged child (10.95 years) in the "challenging" intensity level of the mental condition. The estimated effects of the models and the descriptive statistics of each measure are shown in Figs 6–8. For an easy comparison of the intensity levels and conditions, we cumulated interaction terms with the main effects in the presentations of the results. To calculate the results for the "very easy" and the "very difficult" motor intensity levels, we added the effects of "very easy" and "very difficult", respectively, to the intercept. To calculate the results for the "challenging" motor intensity level, we summed the effects of the "motor" to the intercept. To calculate the results for the "very easy" motor intensity level, we added the effects of "very easy", "motor", and "very easy: motor" to the intercept. To calculate the results for the "very difficult" motor intensity level, we added the effects of "very difficult", "motor", and "very difficult: motor" to the intercept. When we mention increasing or decreasing the load, we always refer to the "challenging" intensity level as the reference.

### 3.1 Game-based control variable: Success rate

The estimated success rate was 90% for the "very easy" mental intensity level, 63% for the "challenging" mental intensity level, and 50% for the "very difficult" mental intensity level. In comparison, it was 100% for the "very easy" motor intensity level, 95% for the "challenging" motor intensity level, and 65% for the "very difficult" motor intensity level. Reducing the load resulted in a significantly increased success rate ($p < 0.001$), while increasing the load led to a significantly decreased success rate ($p < 0.001$) for both conditions. Reducing the load resulted in a significantly smaller increase in the success rate for the motor than for the mental condition ($\Delta = -23\%$, $p < 0.001$). Increasing the load resulted in a significantly larger

**Table 1. Estimated effects for each model and outcome measurement.**

| Outcome | Success Rate (in %) | | | Heart Rate Variability (in RMSSD/RR) | | | Skin Conductance Hand (in µS) | | |
|---|---|---|---|---|---|---|---|---|---|
| **Statistic** | **Estimate** | **P – Value** | **95% CI** | **Estimate** | **P – Value** | **95% CI** | **Estimate** | **P – Value** | **95% CI** |
| Intercept (Mental Challenging) | 62.85 | **<0.001** | 58.08;67.63 | 0.0567 | **<0.001** | 0.0354;0.0873 | 16.70 | **<0.001** | 14.20; 19.20 |
| Age | 0.13 | 0.73 | −0.57;0.82 | −0.0008 | 0.68 | −0.0044;0.0029 | −0.17 | 0.58 | −0.74; 0.40 |
| Very Easy | 27.15 | **<0.001** | 20.81;33.48 | 0.0027 | 0.53 | −0.0056;0.0111 | −0.05 | 0.96 | −2.22; 2.13 |
| Very Difficult | −12.85 | **<0.001** | −19.18;-6.52 | −0.0006 | 0.89 | −0.0089;0.0077 | 0.06 | 0.95 | −2.10;2.25 |
| Motor | 32.06 | **<0.001** | 25.76;38.39 | −0.0148 | **<0.001** | −0.0232;-0.0065 | −0.94 | 0.40 | −3.11;1.23 |
| Very Easy:Motor | −22.52 | **<0.001** | −31.48;-13.57 | 0.0124 | **0.04** | 0.0007;0.0242 | −1.45 | 0.36 | −4.52;1.61 |
| Very Difficult:Motor | −17.47 | **<0.001** | −26.42;-8.52 | −0.0050 | 0.41 | −0.0168;0.0068 | 1.32 | 0.41 | −1.75;4.39 |

| Outcome | Skin Conductance Neck (in µS) | | | %ACmax (in %) | | | %MOVmax (in %) | | |
|---|---|---|---|---|---|---|---|---|---|
| **Statistic** | **Estimate** | **P – Value** | **95% CI** | **Estimate** | **P – Value** | **95% CI** | **Estimate** | **P – Value** | **95% CI** |
| Intercept (Mental Challenging) | 7.43 | **<0.001** | 5.25;9.62 | 24.59 | **<0.001** | 21.33;27.84 | 8.65 | **<0.001** | 7.69;9.61 |
| Age | −0.36 | 0.22 | −0.91; 0.19 | −0.17 | 0.61 | −0.8;0.46 | −0.24 | **0.002** | −0.38;-0.11 |
| Very Easy | −1.09 | *0.10* | −2.34;0.17 | 4 | **0.04** | 0.35;7.65 | 3.39 | **<0.001** | 2.09;4.69 |
| Very Difficult | −0.51 | 0.44 | −1.76;0.75 | −3.26 | *0.09* | −6.91;0.39 | −1.51 | **0.03** | −2.81;-0.21 |
| Motor | −1.29 | **0.05** | −2.55;-0.04 | 52.92 | **<0.001** | 49.27;56.57 | 62.88 | **<0.001** | 61.58;64.18 |
| Very Easy:Motor | 0.84 | 0.38 | −0.94;2.62 | −32.91 | **<0.001** | −38.07;-27.75 | −43.16 | **<0.001** | −45;-41.33 |
| Very Difficult:Motor | 0.54 | 0.59 | −1.24;2.32 | 22.84 | **<0.001** | 17.68;28 | 29.16 | **<0.001** | 27.32;31 |

| Outcome | NASA – Total (#) | | | NASA – Mental (#) | | | NASA – Physical (#) | | |
|---|---|---|---|---|---|---|---|---|---|
| **Statistic** | **Estimate** | **P – Value** | **95% CI** | **Estimate** | **P – Value** | **95% CI** | **Estimate** | **P – Value** | **95% CI** |
| Intercept (Mental Challenging) | 275.00 | **<0.001** | 226.79;323.521 | 58.33 | **<0.001** | 45.45;71.21 | 51.19 | **<0.001** | 39.84;62.54 |
| Age | 7.43 | 0.17 | −2.60;16.69 | 2.21 | 0.11 | −0.38;4.80 | 0.19 | 0.87 | −2.12;2.50 |
| Very Easy | −98.81 | **<0.001** | −150.81;-46.81 | −20.71 | **0.004** | −34.55;-6.88 | −6.91 | 0.27 | −18.92;5.11 |
| Very Difficult | 34.05 | 0.21 | −17.96;86.05 | 10.95 | 0.13 | −2.88;24.79 | −11.19 | *0.08* | −23.21;0.82 |
| Motor | −32.38 | 0.23 | −84.38;19.62 | −17.38 | **0.02** | −31.22;-3.55 | 20.72 | **0.001** | 8.70;32.73 |
| Very Easy:Motor | −6.91 | 0.86 | −80.44;66.63 | 6.43 | 0.53 | −13.14;25.99 | −23.10 | **0.01** | −40.09;-6.10 |
| Very Difficult:Motor | 74.52 | **0.05** | 0.99;148.06 | 0.95 | 0.93 | −18.61;20.52 | 18.81 | **<0.05** | 1.82;35.80 |

| Outcome | NASA – Effort (#) | | | | | | | | |
|---|---|---|---|---|---|---|---|---|---|
| **Statistic** | **Estimate** | **P – Value** | **95% CI** | | | | | | |
| Intercept (Mental Challenging) | 45.95 | **<0.001** | 33.48;58.43 | | | | | | |
| Age | 2.03 | *0.10* | −0.30;4.37 | | | | | | |
| Very Easy | −15.24 | **<0.05** | −29.60;0.87 | | | | | | |
| Very Difficult | 13.33 | *0.08* | −1.03;27.70 | | | | | | |
| Motor | 2.86 | 0.70 | −11.51;17.22 | | | | | | |
| Very Easy:Motor | −6.90 | 0.51 | −27.22;13.41 | | | | | | |
| Very Difficult:Motor | 7.86 | 0.46 | −12.46;28.17 | | | | | | |

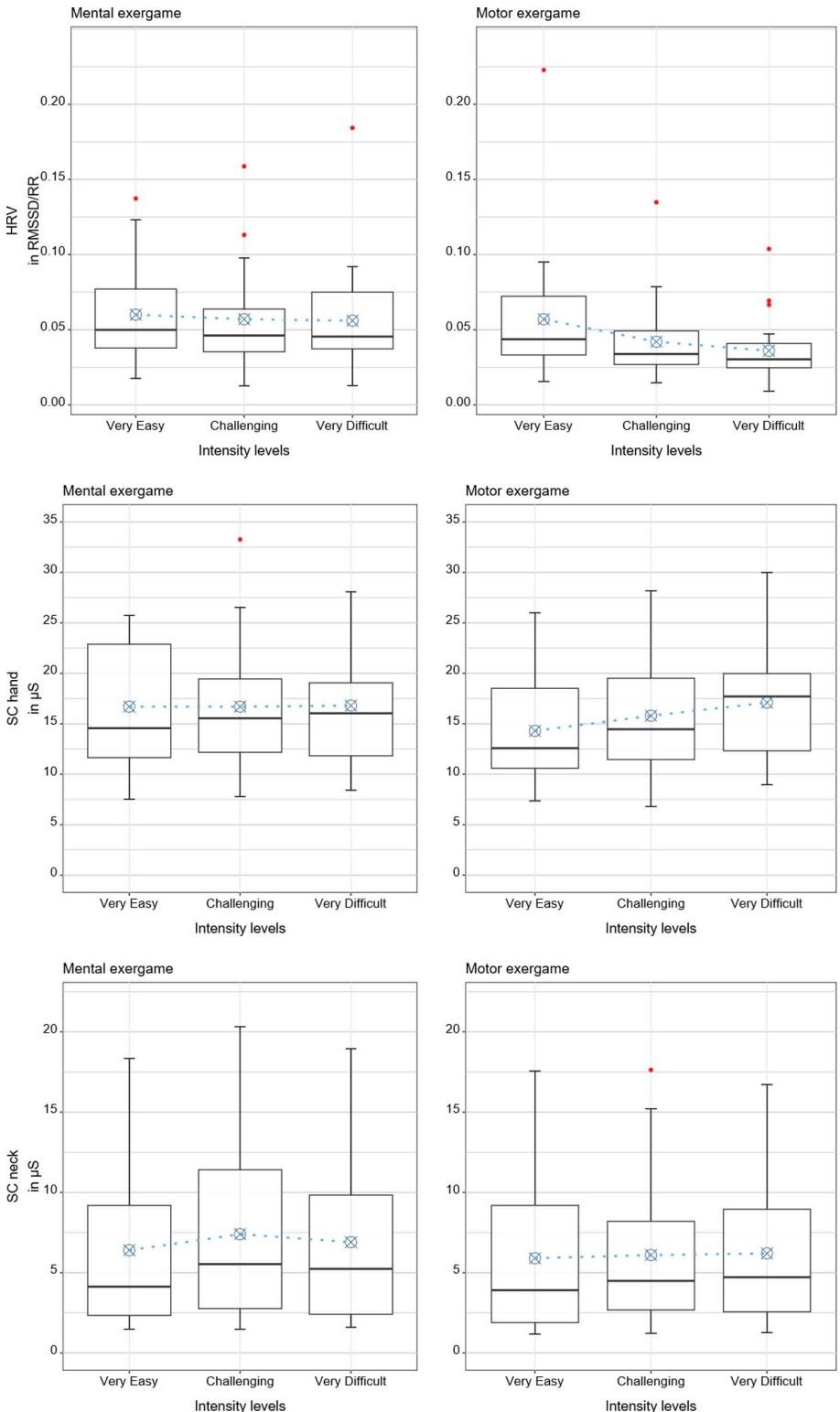

**Fig 6. Results of the physiological measures.** Boxplots with the descriptive statistics, i.e., median and interquartile ranges, and predicted values from the regression models in blue.

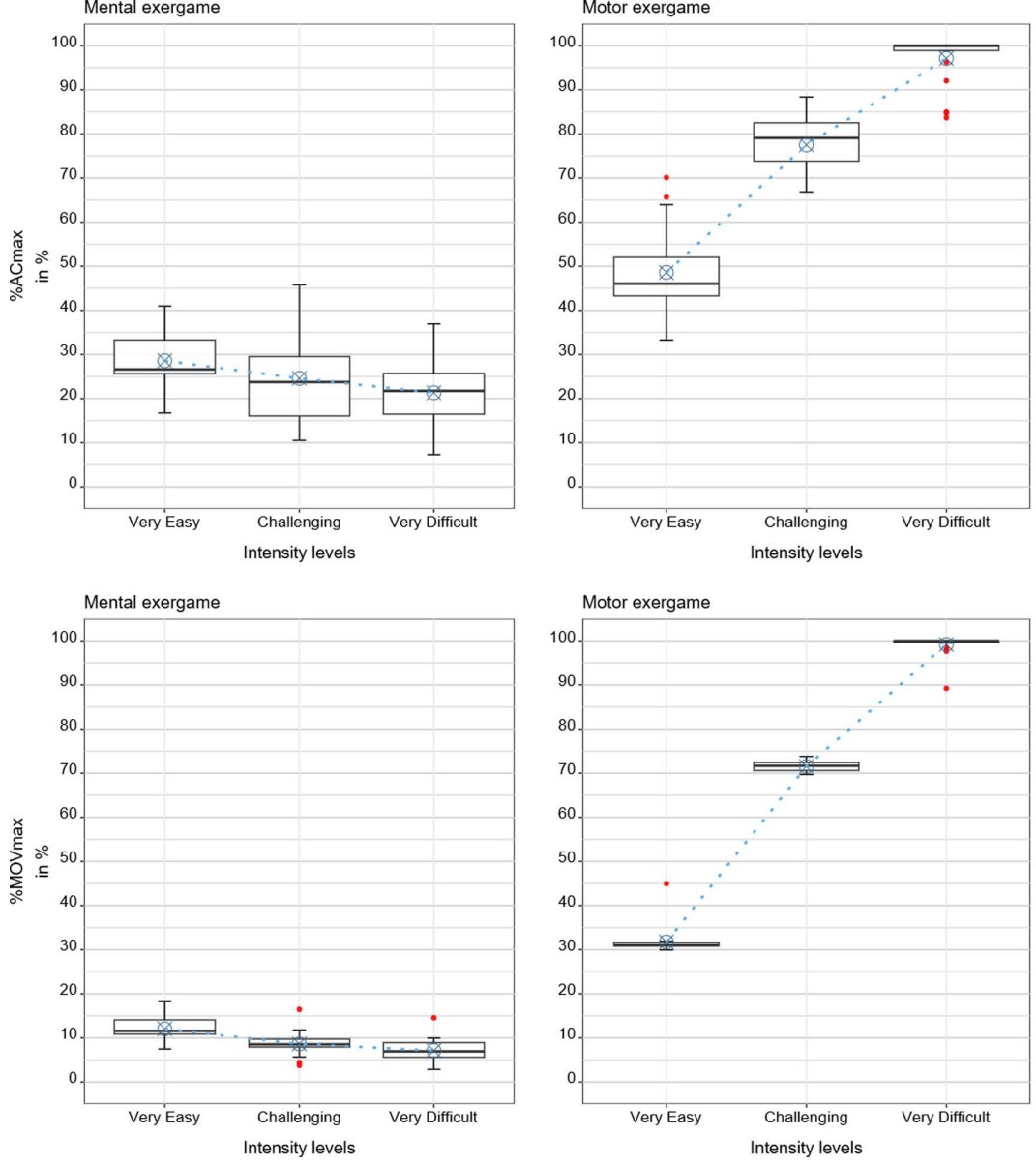

**Fig 7. Results of the movement-based measures.** Boxplots with the descriptive statistics, i.e., median and interquartile ranges, and predicted values from the regression models in blue.

decrease in the success rate for the motor than the mental condition (Δ = −17%, p < 0.001). Age did not significantly affect the success rate (p = 0.73).

### 3.2 Physiological measures: Heart rate variability and skin conductance

The estimated HRV was 0.060 for the "very easy" mental intensity level, 0.057 for the "challenging" mental intensity level, and 0.056 for the "very difficult" mental intensity level, while it was 0.057 for the "very easy" motor intensity level, 0.042 for the "challenging" motor intensity level, and 0.036 for the "very difficult" motor intensity level. Reducing or increasing the load did not significantly affect the HRV for the mental condition. Reducing the load resulted in a significantly larger

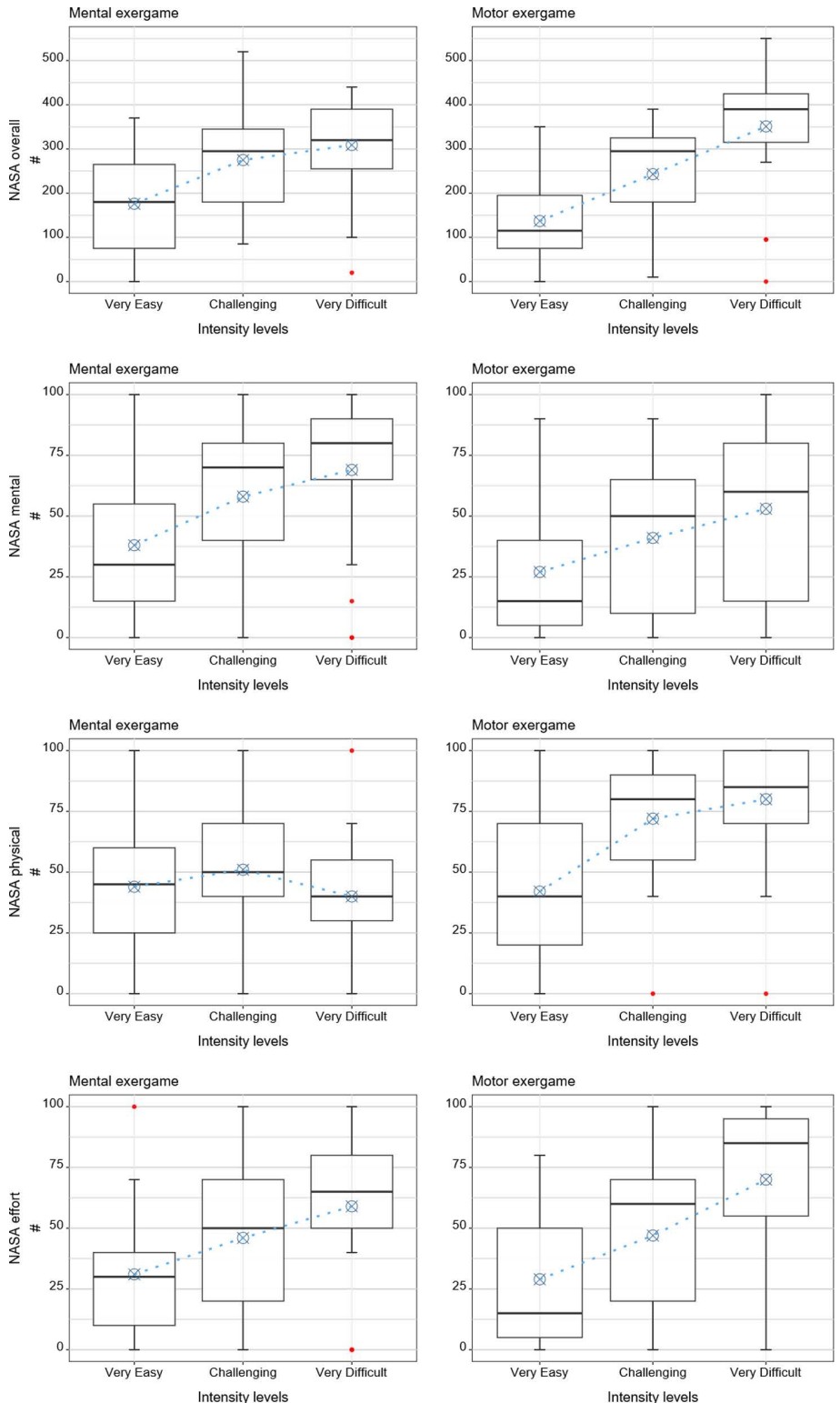

**Fig 8. Results of the patient reported measures.** Boxplots with the descriptive statistics, i.e., median and interquartile ranges, and predicted values from the regression models in blue.

increase in HRV for the motor than for the mental condition ($\Delta = 0.012$, $p = 0.04$). Age did not significantly affect the HRV ($p = 0.68$).

Overall, hand and neck SC were not significantly affected by load or age. Specifically, the estimated hand SC was 16.7 µS for the "very easy" mental intensity level, 16.7 µS for the "challenging" mental intensity level, and 16.8 µS for the "very difficult" mental intensity level. In comparison, it was 14.3 µS for the "very easy" motor intensity level, 15.8 µS for the "challenging" motor intensity level, and 17.1 µS for the "very difficult" motor intensity level. Reducing or increasing the load had no significant effects on the hand SC. Age did not significantly affect the hand SC ($p = 0.58$). Furthermore, the estimated neck SC was 6.4 µS for the "very easy" mental intensity level, 7.4 µS for the "challenging" mental intensity level, and 6.9 µS for the "very difficult" mental intensity level. The estimated neck SC was 5.9 µS for the "very easy" motor intensity level, 6.1 µS for the "challenging" motor intensity level, and 6.2 µS for the "very difficult" motor intensity level. Reducing the load tended to reduce neck SC values for both conditions ($p = 0.1$). Age did not significantly affect the neck SC ($p = 0.22$).

### 3.3 Movement-based measures: Activity counts and number of movement repetitions

The estimated %ACmax was 29% for the "very easy" mental intensity level, 25% for the "challenging" mental intensity level, and 21% for the "very difficult" mental intensity level. In comparison, %ACmax was 49% for the "very easy" motor intensity level, 78% for the "challenging" motor intensity level, and 97% for the "very difficult" motor intensity level. Reducing the load led to significantly higher %ACmax ($p = 0.04$), and increasing the load tended to lower %ACmax ($p = 0.09$) for the mental condition. Reducing the load resulted in a significantly larger decrease in the %ACmax for the motor than for the mental condition ($\Delta = -33\%$, $p < 0.001$). Increasing the load resulted in a significantly larger increase of %ACmax for the motor than for the mental condition ($\Delta = +23\%$, $p < 0.001$). Age did not significantly affect the %ACmax ($p = 0.61$).

The estimated %MOVmax was 12% for the "very easy" mental intensity level, 9% for the "challenging" mental intensity level, and 7% for the "very difficult" mental intensity level. In comparison, it was 32% for the "very easy" motor intensity level, 72% for the "challenging" motor intensity level, and 99% for the "very difficult" motor intensity level. Reducing the load led to a significantly higher %MOVmax ($p < 0.001$), and increasing the load led to a significantly lower %MOVmax ($p = 0.03$) for the mental condition. Reducing the load resulted in a significantly larger decrease in the %MOVmax for the motor than for the mental condition ($\Delta = -43\%$, $p < 0.001$). Increasing the load resulted in a significantly larger increase of the %MOVmax for the motor than for the mental condition ($\Delta = -17\%$, $p < 0.001$). Age significantly affected the %MOVmax ($p = 0.002$), i.e., the %MOVmax at which participants practiced decreased by 0.24% for each year increase from the mean age, and vice versa.

### 3.4 Participant-reported measure: NASA-TLX questionnaire

The estimated NASA-TLX overall score was 176 points for the "very easy" mental intensity level, 275 points for the "challenging" mental intensity level, and 309 points for the "very difficult" mental intensity level. In comparison, it was 137 points for the "very easy" motor intensity level, 243 points for the "challenging" motor intensity level, and 351 points for the "very difficult" motor intensity level. Reducing the load decreased the overall NASA-TLX scores for both conditions significantly ($p = 0.002$). Increasing the load increased the overall NASA-TLX scores significantly more for the motor than for the mental condition ($p = 0.05$). Age did not significantly affect the NASA-TLX overall scores ($p = 0.17$).

The estimated score of the NASA-TLX's mental subscale was 38 points for the "very easy" mental intensity level, 58 points for the "challenging" mental intensity level, and 69 points for the "very difficult" mental intensity level. It was 27 points for the "very easy" motor intensity level, 41 points for the "challenging" motor intensity level, and 53 points for the "very difficult" motor intensity level. Reducing the load decreased the NASA-TLX's mental subscale scores significantly for both conditions ($p = 0.005$), while increasing the load showed no significant effects. Age did not significantly affect the NASA-TLX mental subscale scores ($p = 0.11$).

The estimated score of the NASA-TLX's physical subscale was 44 points for the "very easy" mental intensity level, 51 points for the "challenging" mental intensity level, and 40 points for the "very difficult" mental intensity level. In comparison, it was 42 points for the "very easy" motor intensity level, 72 points for the "challenging" motor intensity level, and 80 points for the "very difficult" motor intensity level. Reducing the load showed no significant effects (p = 0.27), while increasing the load tended to increase the NASA-TLX physical subscale scores on the mental condition (p = 0.08). Reducing the load decreased the scores significantly more for the motor than for the mental condition (Δ = −23, p = 0.01). Increasing the load increased the scores significantly more for the motor than for the mental condition (Δ = +18, p = 0.04). Age did not significantly affect the NASA-TLX physical subscale scores (p = 0.87).

The estimated score of the NASA-TLX's effort subscale was 31 points for the "very easy" mental intensity level, 46 points for the "challenging" mental intensity level, and 59 points for the "very difficult" mental intensity level. It was 27 points for the "very easy" motor intensity level, 49 points for the "challenging" motor intensity level, and 70 points for the "very difficult" motor intensity level. Reducing the load decreased the scores significantly for both conditions (p = 0.043), while increasing the load showed no significant effects. Age tended to affect the NASA-TLX effort subscale (p = 0.10), i.e., the effort subscale scores increased by 2 points for each year increase from the mean age, and vice versa.

## 4 Discussion

This study investigated the responses of several candidate intensity measures to different mental and motor intensity levels of a self-developed upper limb robotic exergame in typically developing children and adolescents. This study indicates that the candidate intensity measures react differently to load type, intensity level, and age in typically developing children and adolescents. The main results were the following: first, the NASA-TLX overall score and its mental and effort subscales increased from the "very easy" to the "challenging" intensity level of the mental condition, and the effort subscale tended to increase from the "challenging" to the "very difficult" intensity level of the mental condition. Second, for the motor condition, %ACmax, %MOVmax, and the NASA-TLX and its physical subscale increased from the "very easy" to the "challenging" and from the "challenging" to the "very difficult" intensity levels. Heart rate variability, the NASA-TLX mental subscale, and the NASA-TLX effort subscale only increased from the "very easy" to the "challenging" intensity level. Third, we only found an effect of age on %MOVmax.

### 4.1 Effects of intensity level and load type

The success rate, our control variable, was the only measure that responded to all intensity levels for each type. We expected such results, as it was our purpose to design two different gaming conditions, i.e., the motor and the mental exergames, each with three intensity levels, i.e., "very easy," "challenging," and "very difficult". These objective data confirm the appropriateness of our task difficulty setting, with the "very easy" intensity level exhibiting the highest success rate and the "very difficult" intensity level the lowest. In addition, the success rate percentages were similar to the ones we targeted, suggesting that we had designed the games adequately and that they have a valid construct.

Heart rate variability decreased at higher motor intensity levels but remained stable at higher mental intensity levels, indicating different physiological responses to each load type, only partially confirming our first hypothesis. The HRV response to higher mental load levels diverges from the existing literature, which reports a decrease in HRV under increased load [46–49]. One explanation for this discrepancy could lie in the nature of the tasks performed by the participants. Previous studies investigating mental load often utilized tasks such as chess or driving, which heavily involve decision-making. Complex decision-making tasks are known to elevate mental load [50], and, therefore, it is possible that our game, lacking decision-making, failed to induce sufficient mental load to elicit changes in HRV. However, decision-making mechanisms are not the only factors that affect mental load. Our game manipulated the number and characteristics of distractors, which also play a crucial role in determining mental load during task completion [51]. This suggests that additional factors beyond the lack of decision-making demands may have influenced our results. One such

factor could be that, as seen in the %MOVmax results, the participants performed at a significantly lower percentage of their MOVmax with increasing mental intensity. The reduced motor load may have interfered with the effects of increased mental load and contributed to the absence of significant HRV differences between mental load levels.

Skin conductance did not respond to increasing levels of motor or mental intensity, so we must reject our initial hypothesis. These results diverge from the existing literature, as SC has been reported to react to changes in task difficulty during video gaming [23,25,52,53]. As outlined by the Publication Recommendations for Electrodermal Recording [54], several factors may have contributed to these insignificant results. Despite our efforts to control environmental variables such as room temperature and humidity, other factors such as skin hydration and electrode detachment remained beyond our control, potentially contributing to the observed insignificant results. Unlike the studies we found in the literature, where the tasks had reduced or inexistent motor load, both our exergames required body activity. This increased body activity made cable management difficult and may have contributed to body hydration, leading to electrode detachment. While tape fixation could have potentially mitigated electrode detachment, it also posed the risk of increasing skin hydration even more by restricting water evaporation.

Percentage ACmax and %MOVmax increased at higher motor intensity levels, which is in line with the existing literature [55] and confirms the first part of our second hypothesis. In contrast, they decreased at higher mental intensity levels, exhibiting opposite responses to each load type, so we must reject the second part of the hypothesis. Activity counts and the number of movement repetitions have always held the potential for becoming intensity measures for upper limb motor rehabilitation. However, in their absolute form, they represent the amount of movement patients perform during therapy, which does not necessarily reflect how hard they work. By normalizing their output to a maximal motor capacity, we obtained information about how hard, in relative terms, the participants worked.

The NASA-TLX overall scores increased significantly from the "very easy" to the "challenging" intensity level for the mental and motor conditions and from the "challenging" to the "very difficult" intensity level for the motor condition, partially confirming our first hypothesis. Laurie-Rose and colleagues [30] suggested the NASA-TLX could be used to differentiate between task load levels in children. They compared two task load levels (i.e., low and hard). When looking at their success rate and NASA-TLX results, it appears that their "low" condition is similar to our "very easy" intensity level and that their "hard" condition is similar to our "challenging" intensity level. Although we did not detect statistically significant differences between the "challenging" and "very difficult" intensity levels for the mental condition, the results indicate the potential of the NASA-TLX to also differentiate between intensity levels at higher intensities.

The analysis of the NASA-TLX physical subscale elucidates that the scores increased throughout the three intensity levels of the motor exergame. In comparison, the mental subscale only significantly increased throughout two intensity levels of the mental exergame. This, together with the results from the overall NASA-TLX implies that the NASA-TLX may function better to when motor demands are involved. Individual responses from participants corroborate these findings, with sixteen participants (76%) successfully scoring the intensity levels incrementally, i.e., they scored the "very easy" intensity levels the lowest, the "challenging" in the middle, and the "very difficult" the highest, while only eight participants (38%) achieved the same categorization for the mental condition.

Although the results only approached significance for the "challenging" - "very difficult" comparison of the NASA-TLX effort subscale, the effort subscale appears to represent overall exercise intensity better than the NASA-TLX overall score. Condensing the questionnaire into only one question could be desirable, particularly in children, if the additional information extracted by the other subscales is unnecessary.

The absence of significant increases from the "challenging" to the "very difficult" intensity level of the NASA-TLX overall score and its subscales for the mental conditions may be attributable to the high data variability. This variability can be ascribed to various factors. For instance, how a task is completed can significantly influence subjective ratings [56]. When the task is completed successfully until the end, it often creates a sense of satisfaction, resulting in a lower perceived workload. Conversely, encountering difficulties in the final phase of a task may evoke frustration and disappointment,

leading to higher perceived workload ratings. In addition, individual characteristics such as sex and personality can play a significant role in the perception of workload due to its multifaceted nature [57]. Moreover, age- and task-related differences in rating behavior have been noted, with younger individuals relying more on extreme ratings than older individuals when rating emotion-based but not physical tasks [58]. These issues complicate the comparisons across age groups and conditions. In within-subject and within-setting comparisons, the workload can be detected with high sensitivity, whereas sensitivity decreases in between-subject comparisons [59], a trend observed in our study. Moreover, the variability of the data poses challenges in classifying workload. Although certain thresholds have been proposed, such as scores above 70 indicating high workload and scores below 30 indicating minimal workload [30], the relative nature of subjective ratings hinders cross-participant comparisons [17]. Establishing generalizable workload classifications for task-specific exercises remains challenging [59], contributing to the lack of standardized data and benchmarks for acceptable workload levels or points of overload [17].

## 4.2 Effects of age

Despite that several studies showed that HRV and SC decrease with age [27,28], we did not find these effects, which leads us to reject our initial hypothesis. We assume that the active nature of our protocol may have covered the effects because HRV and SC measurements are usually performed during rest. In addition, we could think of two underlying mechanisms that may have affected our results: 1) older participants have a reduced HRV and SC, and therefore, for the same relative exercise, they may not be able to show such HRV or SC decreases as the younger participants, and 2) the older participants may be more used to being in stressful situations, exhibiting subtler physiological reactions to the same relative load. Both reasons would lead to an equalization of the HRV and SC levels, masking the anticipated effects of age.

Percentage MOVmax decreased slightly but significantly with age, suggesting that older children trained at lower relative intensity levels than younger children. One explanation could be motivation, i.e., that younger children tried harder. Another reason could be the calibration test. We were under the impression that older participants pushed their limits in the test, which led them to reach scores closer to their real maximal capacity. In contrast, younger children did not push as much, leaving room for improvement. However, when we analyzed the absolute number of MOV for the motor condition, the model indicated that the number of MOV increased significantly (+3.24 [2.43;4.05], p < 0.001) for each additional year. This result can be explained by the development of the nervous system, which extends until the age of 10 years, and the maturation of the body structures, which reach maturity after puberty [60], which enables humans to progressively become stronger and faster and to perform more movement repetitions in our game. The difference between the absolute and relative results heightens the importance of measuring intensity in relative manners.

However, in contrast to the results of the absolute or relative MOV, we did not find an effect of age on %ACmax. Even when investigating the effect of age on the absolute AC, it was far from being significant (0.5 [−1.71;2.71], p = 0.67). Activity counts are calculated from three-axial acceleration data and are affected by movement speed and range of motion. Based on this premise, the most logical explanation for the insignificant effects would be that younger participants performed fewer but faster movements. However, this can be refuted by the nature of our assessment, which is based on speed. Therefore, the only plausible explanation lies in the algorithm used to calculate the AC. It may be possible that the algorithm's precision decreases above a certain movement threshold, diluting the potential effects of age.

## 4.3 Potential applicability

Although this is just a first study in typically developing children, the results provide some preliminary information on the potential applicability of the measures for therapy purposes.

Heart rate variability reacted to increasing motor load, showing its potential as an intensity measure for motor tasks. Heart rate variability could be used in the same way heart rate is used in cardiovascular training. For example, we could use HRV to target a desired intensity range online, i.e., while the therapy is ongoing.

We tried identifying the threshold HRV values between two consecutive motor intensity conditions, i.e., "Very Easy" versus "Challenging" and "Challenging" versus "Very Difficult", using Receiver Operator Characteristics (ROC) analyses. We identified the threshold HRV value as the value with the highest combined sensitivity and specificity, i.e., with the highest Youden Index.

The ROC analysis showed that a HRV of 0.035 differentiated best between the "Very Easy" and "Challenging" conditions, with a sensitivity of 71% and specificity of 57%, correctly classifying 64% of the cases (AUC: 0.61, p = 0.04). A HRV of 0.033 differentiated best between the "Challenging" and "Very Difficult" conditions, with a sensitivity of 66% and specificity of 57%, correctly classifying 62% of the cases (AUC: 0.6, p = 0.08). Although the models performed better than a random classifier, their performance was poor.

The two HRV threshold values lie close together, leaving a narrow range of HRV values indicating "challenging" intensity. The narrow range of HRV values might be explained by the lack of normalizing the HRV values for each participant individually. Therefore, within each participant, we normalized the HRV values of each motor intensity level to the HRV values from the maximal motor capacity test and repeated the ROC analyses. These ROC analyses showed that a normalized HRV of 1.754 differentiated best between the "Very Easy" and "Challenging" conditions, with a sensitivity of 62% and specificity of 81%, correctly classifying 71% of the cases (AUC: 0.72, p = 0.004). A normalized HRV of 1.062 differentiated best between the "Challenging" and "Very Difficult" conditions, with a sensitivity of 43% and specificity of 86%, correctly classifying 64% of the cases (AUC: 0.62, p = 0.04).

The combined sensitivity and specificity, and the AUC and p-values were somewhat better for the normalized HRV analyses than the non-normalized HRV analyses, indicating that the normalization accounts for inter-individual variability. However, using HRV in its actual form to adapt therapy intensity remains difficult, as we do not know how these numbers reflect the participant's capacity. Consequently, further efforts are needed so that HRV can be used as an online intensity measure.

Skin conductance does not appear to be a good measure for clinical practice. On the one hand, it did not react to changes in load, and on the other, it shows practicability issues such as cable management or detachment due to sweat.

%ACmax and %MOVmax reacted to increasing motor load, showing potential as an intensity measure for motor tasks. Similarly to the normalized HRV values, %ACmax and %MOVmax account for the participant's maximal capacity. However, the %ACmax and %MOVmax results allow an easier interpretation. While they allow us to know the exact relative intensity at which the participants engage in our motor condition, it could be difficult to extrapolate these findings to other exercises, as it might not always be feasible to count the number of movement repetitions or perform a maximal capacity test for every type of movement, especially in conventional therapy settings. In addition, as the intensity can only be computed once the participant finishes playing the level, these outcomes cannot measure intensity online. Thus, they cannot be used to adapt therapy intensity ongoingly.

The NASA-TLX and its subscales showed significant or near to significant differences for most of the comparison levels, showing potential as intensity measures. Especially the effort subscale could be used as a simplified version of the whole questionnaire if confirmatory studies in patients support our preliminary findings. Furthermore, although it would not allow online measurement of therapy intensity, having to answer only one question could serve as a pseudo-online measure of intensity due to its simplicity and reduced time cost.

### 4.4 Limitations

Overall, mental intensity appeared more challenging to capture than motor intensity. One explanation could be that the differences between the mental intensity levels were too subtle, specifically between the "challenging" and "very difficult" levels. Another reason could be that the participants still had to perform arm movements in the mental condition, which may have masked the effects. For a follow-up study, we recommend adjusting the mental game condition to reduce or completely cancel arm movements. Furthermore, we should take the participants' changing gameplay strategies into account. Cognitive processes remain complex, individual, and invisible, making them difficult to understand.

In addition, the present study focused on two specific exergames, and the participants were typically developing children and adolescents. The results are specific to these games, participants, and study settings. Future studies with larger cohorts of the actual target group, i.e., children and adolescents with neurological diagnoses undergoing upper limb neurorehabilitation, e.g., occupational therapy, could confirm our findings in an ecologically valid way. Future studies should also assess the psychometric properties, e.g., the test-retest reliability of these candidate intensity measures in the target group.

Furthermore, the randomization of the intensity levels may have affected the NASA-TLX results. This is because the task sequence can influence workload ratings [59]: easier tasks are perceived as more difficult than they objectively would be and vice versa. Although this may have affected our results at an individual level, the blocked random order should have compensated for these effects at the group level.

Selecting different percentages for the intensity levels may also have resulted in different results. For instance, if we had set the "challenging" levels at a slightly lower percentage, we may have seen statistically significant differences between certain "challenging" and "very difficult" comparisons, as they approach already, in many cases, significance with the actual percentages.

We used the output data from the motor calibration test as the maximal motor capability. The maximal test allowed us to infer the percentage of the maximum capacity, i.e., the intensity, at which the participants were exercising at each level. As the calibration only lasted thirty seconds, and the different game levels three minutes, we expected that our assessment would overestimate the maximal capacity when extrapolating it to three minutes, and that the intensity percentages of each intensity level would be slightly below what they should be in reality. However, the opposite happened, and several participants (10 for %ACmax; mean = 110%, and 15 for %MOVmax; mean = 103%) slightly surpassed the 100% intensity limit in the "very difficult" condition. The higher performance during the measurements may have to do with the difference in game design between the calibration and the gaming condition. For the calibration, the participants selected the speed and relied on intrinsic motivation to perform at their maximum capacity. For the measurement, they had to match the game requirements, relying on extrinsic motivation. Extrinsic motivation is known to deliver better results for short-term goals and activities [61], which helps explain why %ACmax and %MOVmax were slightly higher in the "very difficult" intensity levels than in the calibration test. Nevertheless, in our opinion, such an approach is needed to convert a measure indicating the amount of movement into a measure reflecting intensity.

## 5 Conclusion

In rehabilitation, accurately measuring therapy intensity is critical in personalizing therapies. We showed with this study that the several candidate intensity measures reacted differently to increases in mental and motor intensity and that measuring mental intensity with objective measures remains complicated. Combining intensity measures may be the best strategy for multidimensionally assessing therapy intensity as each measure has strengths and weaknesses. While we would currently recommend a combination of HRV, %ACmax, and the NASA-TLX effort subscale, we need to verify the current findings and determine the psychometric properties of the various intensity measures in the target group, i.e., pediatric patients with diverse neurological conditions and abilities. As the healthcare landscape continues to evolve, a comprehensive understanding of therapy intensity and its measures will undoubtedly enhance the individualization of therapy, improving its effectiveness and will enable more precise dose-response calculations.

## Acknowledgments

We would like to thank all the participants and their families, as well as to our colleagues, especially to Andrina Kläy, who developed the two exergames.

## Author contributions

**Conceptualization:** Gaizka Goikoetxea-Sotelo, Hubertus J.A. van Hedel.

**Data curation:** Gaizka Goikoetxea-Sotelo, Livia Rätzo.

**Formal analysis:** Gaizka Goikoetxea-Sotelo.

**Funding acquisition:** Gaizka Goikoetxea-Sotelo, Hubertus J.A. van Hedel.

**Investigation:** Gaizka Goikoetxea-Sotelo, Livia Rätzo.

**Methodology:** Gaizka Goikoetxea-Sotelo, Hubertus J.A. van Hedel.

**Project administration:** Hubertus J.A. van Hedel.

**Software:** Gaizka Goikoetxea-Sotelo, Livia Rätzo.

**Supervision:** Hubertus J.A. van Hedel.

**Visualization:** Gaizka Goikoetxea-Sotelo, Livia Rätzo.

**Writing – original draft:** Gaizka Goikoetxea-Sotelo, Livia Rätzo.

**Writing – review & editing:** Gaizka Goikoetxea-Sotelo, Hubertus J.A. van Hedel.

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
