## [Decision Letter · Decision Letter 0]

Dear Dr. Goikoetxea-Sotelo,

Thank you for submitting your manuscript to PLOS ONE. After careful consideration, we feel that it has merit but does not fully meet PLOS ONE’s publication criteria as it currently stands. Therefore, we invite you to submit a revised version of the manuscript that addresses the points raised during the review process.

We look forward to receiving your revised manuscript.

Kind regards,

Yi Ding

Anhui Polytechnic University

Academic Editor

PLOS ONE

Journal Requirements:

“This work is supported from grants from the J&K Wonderland Foundation and the Anna-Müller Grocholski Foundation. The funders were not involved at any stage of this research”

4. We note that you have indicated that there are restrictions to data sharing for this study. PLOS only allows data to be available upon request if there are legal or ethical restrictions on sharing data publicly. For more information on unacceptable data access restrictions, please see http://journals.plos.org/plosone/s/data-availability#loc-unacceptable-data-access-restrictions .  

Reviewers' comments:

Reviewer's Responses to Questions

**Comments to the Author**

1. Is the manuscript technically sound, and do the data support the conclusions?

Reviewer #1: Partly

2. Has the statistical analysis been performed appropriately and rigorously?

Reviewer #1: No

3. Have the authors made all data underlying the findings in their manuscript fully available?

Reviewer #1: Yes

4. Is the manuscript presented in an intelligible fashion and written in standard English?

Reviewer #1: Yes

Reviewer #1: The purpose of this is study is to investigate the response of physiological variables such as heart rate variability (HRV), skin conductance (SC), activity counts and movement repetitions normalized for the maximal capacity (%ACmax and %MOVmax, respectively), and cognitive variables as the NASA-TLX scale to different mental and motor intensity levels of two self-developed upper limb exergames in typically developing children. The authors also investigated the effects of age on the responses of the measures.

The article is interesting, but the relevance and applicability is not very clear. Here are my suggestions:

- The purpose should be clearly stated, avoiding excessive use of acronyms.

- The purpose in the abstract differs from the one presented in the main text.

Introduction:

- The introduction provides detailed information on different types of therapy and their outcomes. However, there is no mention or explanation regarding exergames in the introduction, or how the results from this study can help the therapy procedures.

- The significance of exergames and their relevance to the study's purpose should be addressed. It is unclear how exergames influence the therapy in question.

Methods:

Figure 1 please does not identify the patient.

On page 9, reference number should be added for line 205.

The sample rate for the Polar system needs to be described.

Additionally, the heart rate units used are uncommon and make comparisons with other studies challenging.

Provide more details about the NASA TLX questionnaire, including at least one more paragraph.

Results:

Variables in Table 1 should include their corresponding units.

Figures 6 and 7 are difficult to interpret due to poor resolution.

General comments:

The sample size is small, especially when divided into three age groups.

Based on motor development, the sample size is insufficient to support strong conclusions.

Lastly, the discussion does not clearly explain the relationship between exergames and the therapy.

The study can be very interesting, but does not show clearly its applicability for therapy purposes. For example, what is the importance of HRV increase or decrease in therapy? How 3min gaming practice can improve children skills in the long term?

I hope the comments are useful and help the reader to understand the work more clearly.

**Do you want your identity to be public for this peer review?** For information about this choice, including consent withdrawal, please see our Privacy Policy

Reviewer #1: No

---

## [Author Response · Author response to Decision Letter 1]

6 Jan 2025

Dear Editor,

We adapted the points 1-4 as follows:

1. We adapted the style of the manuscript and title page based on the recommendations and PLOS ONE guidelines.

2. We made the game available in the Harvard Dataverse repository. In this way, the study can be replicated.

3. We included the required funding information in the cover letter as suggested.

4. We uploaded the anonymized data to the Harvard Dataverse repository together with the exergame data as requested.

The response to the reviewer can be found in a separate document that we uploaded together with the manuscripts.

If there is anything missing, please let us know.

Beste regards,

Gaizka

---

## [Decision Letter · Decision Letter 1]

Dear Dr. Goikoetxea-Sotelo,

We look forward to receiving your revised manuscript.

Kind regards,

Joanna Tindall, PhD

Staff Editor

PLOS ONE

Journal Requirements:

Additional Editor Comments (if provided):

Reviewers' comments:

Reviewer's Responses to Questions

**Comments to the Author**

Reviewer #1: All comments have been addressed

2. Is the manuscript technically sound, and do the data support the conclusions?

Reviewer #1: Yes

3. Has the statistical analysis been performed appropriately and rigorously?

Reviewer #1: Yes

4. Have the authors made all data underlying the findings in their manuscript fully available?

Reviewer #1: Yes

5. Is the manuscript presented in an intelligible fashion and written in standard English?

Reviewer #1: Yes

Reviewer #1: I would like to thank the authors for the very detailed answers on the response to my comments. It is much more clear now the purpose of the work. The figure added in the review is very clear as well.

some minor comments:

- line 69, the sentence does not look complete.

- Although mentioned the NASA-TLX questionnaire as a supplementary material, in the pdf I have access, it does not show. I suggest the authors to upload it in a repository, same as you did with the games version.

**Do you want your identity to be public for this peer review?** For information about this choice, including consent withdrawal, please see our Privacy Policy

Reviewer #1: **Yes: ** Denise Paschoal Soares

---

## [Author Response · Author response to Decision Letter 2]

7 May 2025

Reviewer #1:

- Comment: I would like to thank the authors for the very detailed answers on the response to my comments. It is much more clear now the purpose of the work. The figure added in the review is very clear as well.

Answer: We would like to thank the reviewer for the previous and this last remarks, which we consider have improved the quality of our manuscript.

some minor comments:

- Comment: Line 69, the sentence does not look complete.

Answer: We have adapted this sentence for clarity.

- Comment: Although mentioned the NASA-TLX questionnaire as a supplementary material, in the pdf I have access, it does not show. I suggest the authors to upload it in a repository, same as you did with the games version.

Answer: We suppose this is an error from the system, as we can see the document in the uploads. We have, however, added the NASA-TLX document to the repository as suggested.

---

## [Decision Letter · Decision Letter 2]

Responses of candidate intensity measures to different mental and motor load levels using upper limb exergames in typically developing children and adolescents

PONE-D-24-29809R2

Dear Dr. Goikoetxea-Sotelo,

We’re pleased to inform you that your manuscript has been judged scientifically suitable for publication and will be formally accepted for publication once it meets all outstanding technical requirements.

Kind regards,

Steven J. Landry, Ph.D.

Academic Editor

PLOS ONE

Additional Editor Comments (optional):

Thank you for your submission and your careful consideration of the reviewers' comments.

Reviewers' comments:

Reviewer's Responses to Questions

**Comments to the Author**

Reviewer #1: All comments have been addressed

2. Is the manuscript technically sound, and do the data support the conclusions?

Reviewer #1: Yes

3. Has the statistical analysis been performed appropriately and rigorously?

Reviewer #1: Yes

4. Have the authors made all data underlying the findings in their manuscript fully available?

Reviewer #1: Yes

5. Is the manuscript presented in an intelligible fashion and written in standard English?

Reviewer #1: Yes

Reviewer #1: I would like to thank the authors for the reviewed version. The paper looks very good.

Wish you the best with your future publications.

**Do you want your identity to be public for this peer review?** For information about this choice, including consent withdrawal, please see our Privacy Policy

Reviewer #1: **Yes: ** Denise Soares

---

## [Editor Report · Acceptance letter]

PONE-D-24-29809R2

PLOS ONE

Dear Dr. Goikoetxea-Sotelo,

I'm pleased to inform you that your manuscript has been deemed suitable for publication in PLOS ONE. Congratulations! Your manuscript is now being handed over to our production team.

Kind regards,

on behalf of

Dr. Steven J. Landry

Academic Editor

PLOS ONE